# Nano-Silicon@Exfoliated Graphite/Pyrolytic Polyaniline Composite of a High-Performance Cathode for Lithium Storage

**DOI:** 10.3390/ma16041584

**Published:** 2023-02-14

**Authors:** Qian Wu, Yinghong Zhu, Haojie Duan, Lin Zhu, Yuting Zhang, Hongqiang Xu, Ishioma Laurene Egun, Haiyong He

**Affiliations:** 1College of Chemical Engineering, Zhejiang University of Technology, Hangzhou 310014, China; 2Ningbo Institute of Materials Technology and Engineering, Chinese Academy of Sciences, Ningbo 315201, China; 3Department of Chemical and Environmental Engineering, Faculty of Science and Engineering, University of Nottingham Ningbo China, Ningbo 315100, China

**Keywords:** lithium-ion batteries, anode material, polyaniline pyrolysis, exfoliated graphite

## Abstract

In this paper, a Si@EG composite was prepared by liquid phase mixing and the elevated temperature solid phase method, while polyaniline was synthesized by the in situ chemical polymerization of aniline monomer to coat the surface of nano-silicon and exfoliated graphite composites (Si@EG). Pyrolytic polyaniline (p-PANI) coating prevents the agglomeration of silicon nanoparticles, forming a good conductive network that effectively alleviates the volume expansion effect of silicon electrodes. SEM, TEM, XRD, Raman, TGA and BET were used to observe the morphology and analyze the structure of the samples. The electrochemical properties of the materials were tested by the constant current charge discharge and cyclic voltammetry (CV) methods. The results show that Si@EG@p-PANI not only inhibits the agglomeration between silicon nanoparticles and forms a good conductive network but also uses the outermost layer of p-PANI carbon coating to effectively alleviate the volume expansion of silicon nanoparticles during cycling. Si@EG@p-PANI had a high initial specific capacity of 1491 mAh g^−1^ and still maintains 752 mAh g^−1^ after 100 cycles at 100 mA g^−1^, which shows that it possesses excellent electrochemical stability and reversibility.

## 1. Introduction

The negative electrode is the transfer medium of lithium ions and electrons in the process of battery charging [1,2] and plays the role of charge storage and release. Concerning battery costs, the negative electrode material accounts for about 5–15%, thus making it one of the key components of lithium-ion batteries (LIB). The requisite for LIBs is not only a long lifespan and environmental friendliness but also a high energy/power density [3]. It is imperative to develop anode materials with a larger capacity, because the capacity of the traditional graphite anode has been developed to the limit [4,5,6]. Silicon-based anode material is one of the materials with good application prospects at present. The theoretical capacity of the silicon anode is about ten times that of graphite, because one silicon atom can combine with four lithium ions [7,8]. In addition, silicon has rich reserves, a low cost, a low toxicity and environmental friendliness [9,10,11]. However, it should be noted that the capacity of the silicon anode decreases rapidly because it expands greatly during the discharging cycle, which also leads to the active material falling off the copper foil of the current collector, resulting in a loss of electrical contact [12,13,14,15]. Furthermore, the low conductivity of silicon itself has a bad influence on the battery performance [16]. Therefore, completely replacing traditional graphite with an Si electrode is still unrealistic [17,18].

In recent years, many strategies have been proposed to address these shortcomings of silicon. First, by reducing the size of silicon particles, it is found that there is a critical particle size of about 150 nm for the lithification of single silicon nanoparticles. When the critical particle size is lower, the particles will not break during the first lithification [19]. This nanoscale can alleviate part of the volume expansion. Furthermore, compounding with different materials can further alleviate the volume expansion and improve the ionic conductivity. Silicon nanoparticles are coated with conductive carbon and polymer protective coating, which is similar to three-dimensional composite or hollow spherical carbon material [20,21,22,23,24,25]. These structures have certain advantages, because their internal voids and spaces will prevent the expansion of silicon to a certain extent, while the outer carbon layer and conductive polymer can be used as fast transmission channels for electrons and ions.

In this paper, we prepared a kind of composite material, which first coated silicon nanoparticles with exfoliated graphite; the outermost layer that is coated is further subjected to polyaniline pyrolysis to obtain a nitrogen-doped carbon layer. The preparation method is simple and easy to obtain. In this structure, the silicon nanoparticles are sandwiched in between the exfoliated graphite layer, and the nitrogen-doped carbon obtained by the high-temperature pyrolysis of polyaniline is used as the adhesive. This unique structure makes the composite structure more stable and forms an interconnected conductive network. This ensures that the silicon nanoparticles have enough buffer space during the electrochemical cycle process, so they have a stable long-term electrochemical cycle performance. The three-dimensional structure material shows a highly reversible capacity of 1100 mAh g^−1^ in the first ten cycles, and the specific capacity remains at 752.9 mAh g^−1^ after 100 cycles.

## 2. Materials and Methods

The preparation of the exfoliated graphite-coated Si nanoparticles composite (Si@EG): To obtain the Si@EG, 1 g of exfoliated graphite (self-made, purity: 99.9%, size: 5–6 μm) was first dispersed in 200 mL of deionized water under ultrasonication for 30 min and was subjected to magnetic stirring for 120 min. Subsequently, Si nanoparticle powder (commercial, size: 100–150 nm) was slowly added into the above-mentioned solution under ultrasonication for 2 h, followed by stirring for 2 h. The resultant Si@EG sample was obtained by the freeze-dried sample, and it was calcined at 650 °C in an argon atmosphere for 7 h.

The preparation of the Si@EG@p-PANI composite: 200 mg of the Si@EG composite was ultrasonically dispersed into 200 mL of deionized water. The hydrochloric (200 mg, AR: 98%) aqueous solution and aniline monomer (AR: 99.5%) (molar mass 1:20) were dripped into the above suspension. After that, a certain amount of ammonium persulfate (APS, AR: 99.99%) oxidant was dissolved into 40 mL of 1 M HCL solution that was precooled in advance, and then it was added into the abovementioned mixed solution. The molar mass ratio of APS/aniline is 3:2, and it is continuously stirred in an ice-water bath for 24 h until the color of the solution changes to dark green. The dark green product was filtered and washed three times with ethanol, and the obtained filter cake was dried in a vacuum oven at 60 °C for 12 h. The resultant product was denoted as Si@EG@p-PANI after being calcined at 700 °C in an argon atmosphere for 7 h.

Materials Characterizations: The surface morphology of the samples was investigated by an S-4800 cold field emission scanning electron microscope (HITACHI, Japan). High-resolution images of the samples were obtained by transmission electron microscopy (TEM, Tecnai F 0, HITACHI, Japan). The crystallographic structure of the as-synthesized samples was characterized by X-ray diffractometry (XRD, Bruker, Germany) (D8 Advance Davinci) with Cu Kα radiation (λ = 0.15406 nm) at a scanning range of 5–90°, with a scanning speed of 6° min^−1^. Raman spectra (inVia-reflex) were obtained on a Jobin-Yvon LabRAM HR Evolution spectrometer (Renishaw, UK) equipped with a 532 nm laser. Thermogravimetric analysis (TGA) was carried out using a TGA 8000-Spectrum two-Clarus SQ8T instrument (Perkin Elmer, US) at a heating rate of 10 °C min^−1^ from 20 °C to 900 °C under air. The Brunauer Emmett Teller (BET) specific surface area was analyzed by nitrogen adsorption on a Micromeritics ASAP 2020M nitrogen adsorption apparatus (Micrometric, USA). X-ray photoelectron spectroscopy (XPS, AXIS ULTRA DLD) was performed on a Thermo Scientific K-Alpha spectrometer (Shimadzu, Japan) equipped with a monochromatic Al Kα X-ray source (hν = 1486.6 eV).

Electrochemical Measurements of the Half-Cell Testing: The first step is to prepare the electrode slurry, fully mix the active material, conductive carbon (Super P) and adhesive sodium carboxymethyl cellulose (CMC) according to the mass ratio of 7:1:2 and mix them in the slurry bottle with deionized water as the solvent for about 6 h to make them fully uniform. The slurry is then uniformly coated on the copper foil and dried under vacuum at 80 °C for 12 h. Then, the sheet punch is used to cut the electrode sheet with a diameter of 12 mm. Then, using lithium metal foil as the counter electrode, the CR2032 half-cell was assembled in a glove box filled with argon without moisture, and 1 M LiPF6 was dissolved in ethylene carbonate (EC)/diethyl carbonate (DEC)/ethylene methyl carbonate (EMC) (1:1:1, volume ratio) as the electrolyte.

The multichannel battery test system (LAND CT-2001A, Wuhan Rambo Testing Equipment Co., Ltd., WuHan, China) was applied to measure the galvanostatic charge–discharge in the voltage range from 0.01 to 3 V (vs. Li/Li^+^) and at different current densities. The cyclic voltammetry (CV) curves of the electrode are obtained by using a Solartron 1470E (Solartron Public Co., Ltd., ShangHai, China) multi-channel potentiostats electrochemical workstation in the voltage range of 0.01–3 V (vs. Li/Li^+^) at a scanning rate of 0.1 mV s^−1^. All the electrochemical experiments indicated above were conducted at room temperature.

## 3. Results

The morphology and the microstructure of the two samples of Si@EG and Si@EG@p-PANI were characterized by SEM and TEM images, as shown in Figure 1 and Figure 2, respectively. The outmost exfoliated graphite can be clearly observed (Figure 1b). The Si@EG shows an ultrathin two-dimensional multi-graphite sheet (Figure 2b) and silicon nanoparticles embedded in it. It is conducive to the uniform dispersion of silicon nanoparticles in the interlayer of graphite (Figure 1b and Figure 2a). The thin-layer structure of stripped graphite gives it the properties of curl, wrinkle and flexibility, which can effectively buffer the volume change of the silicon negative electrode material in the process of charge and discharge [26]. However, the SEM results showed a denser coating sample when p-PANI was added (Figure 1d). For Si@EG@p-PANI composites (Figure 2c,d), crystalline silicon nanoparticles and nitrogen-doped carbon after the carbonization of polyaniline can be observed. The carbonized particles of polyaniline are on the uniformly distributed conductive matrix.

The X-ray diffraction (XRD) patterns of Si, Si@EG and Si@EG@p-PANI are shown in Figure 3a. Si, Si@EG and Si@EG@p-PANI have the same characteristic peaks at 2θ = 28.44°, 47.30° and 56.12°, which correspond to the Si (111), (220), and (311) crystal surface (JCPDS27-1402) [27]. This result is consistent with the results of the SEM and TEM, which show that Si successfully exists in the composite material and that there is no structural change in Si during the process of synthesizing the composite material. In the Si@EG composites, the characteristic peak of EG also appears near 26.6°, and its peak intensity is stronger than that of Si. It is worth noting that the amorphous carbon formed with the pyrolysis of polyaniline reduces the overall order of the composites, so the crystalline strength of silicon increases accordingly. Moreover, the grain size of silicon was calculated using Equation (1) from the data of XRD—a value of 0.1274 nm. In Equation (1), “K” represents the Scherrer constant, “µ” represents the X-Ray Wavelength, “β” represents the integral half-width and “θ” denotes the diffraction angle. The relative crystallinity is 85%, which is calculated using Equation (2). In Equation (2), “Ic” indicates the integral strength of the crystallization peak and “Ia” represents amorphous peaks [28]. The XRD result (Figure 3e) further confirms the crystalline nature of Si, EG and p-PANI in Si@EG@p-PANI.
(1)D=Kμβcosθ
(2)∈=IcIc+Ia

Figure 3b shows the Raman spectroscopy results of Si, Si@EG and Si@EG@p-PANI to further elucidate the microstructure of amorphous carbon. As shown in Figure 3b, the characteristic Raman mode of crystal silicon presents the same characteristic peak near 515 cm^−1^ [29]. As shown in Figure 3b, the Raman characteristic peak intensity of silicon is very strong in the composite Si@EG, which may be due to the materials being unevenly mixed. Except for silicon, two characteristic peaks were detected at 1348 cm^−1^ (D band) and 1580 cm^−1^ (G band). The D band corresponds to the structural defects and disordered structure in graphite, while the G band indicates the stretching vibration of the bond between the sp^2^ atoms. With the increase in the amorphous carbon, the intensity of the G peak is weakened, and the peak shape is gradually widened [30,31]. The ID/IG ratios of Si@EG, and Si@EG@p-PANI are 1.15, and 2.55, respectively. It shows that amorphous carbon particles are added on the graphite surface. The Raman signal is mixed, and the phonons are influenced by the carbon particles, showing a higher D. The PANI heat treatment leads to an increased disorder of Si@EG@p-PANI. Therefore, adding p-PANI will weaken the G band and strengthen the D band, which will increase the ID/IG value.

The mass ratio of Si in Si@EG and Si@EG@p-PANI is investigated by thermogravimetric analysis (Figure 3c). Due to the oxidation of silicon, there is a slight increase in the mass of silicon (1.92%). Additionally, from the TGA curve, the major weight loss of the Si@EG composite occurs between 600 °C and 800 °C. This is due to the rapid oxidation of EG, with a mass loss of 76.78%. The mass ratio of Si@EG can be estimated (25.14 wt% of silicon and 74.86 wt% of exfoliated graphite). The partial weight loss above 400 °C may be due to the presence of water molecules and some non-carbonized polymer molecular chains [32]. Furthermore, the thermal breakdown of Si@EG@p-PANI composites occurs between 400 °C and 610 °C, When the temperature reached about 400 °C, the Si@EG@p-PANI composites began to have a significant mass loss, which was due to the decomposition of carbon and nitrogen in the composites. Through the TGA curves, the mass ratio of Si@EG@p-PANI can be calculated (15.91 wt% of silicon and 84.09 wt% of EG and p-PANI).

Figure 3d presents the nitrogen sorption isotherms of Si, Si@EG and Si@EG@p-PANI and the homologous distributions of the pore size. The specific surface area of Si is 21.2121 m^2^ g^−1^, that of Si@EG is 26.7436 m^2^ g^−1^ and that of Si@EG@p-PANI is 71.0018 m^2^ g^−1^. The corresponding pore size distributions are calculated by the Barrett–Joyner–Halenda model. The pore size distribution curve of Si@EG (Figure 3f) shows that the main pore size is 3.68 nm, the total pore volume is 0.0092 cm^3^ g^−1^ nm and the wide pore size range is 1.9–17 nm, indicating that most of the pores are mesopores. After coating p-PANI, the hysteresis loop almost disappears, and the total pore volume decreases to 0.0018 cm^3^ g^−1^ nm. These results imply that the specific surface area of Si@EG@p-PANI is higher than that of Si@EG. This is because p-PANI has formed an abundant nanoscale pore structure.

X-ray photo electron spectroscopy (XPS) was employed to further investigate the surface composition and the chemical state of each element for the Si@EG@p-PANI and Si@EG samples. Figure 4a,e show the whole XPS spectra of Si@EG@p-PANI and the Si, C, O and N elements, while the Si@EG samples show the Si, C and O elements. The Si@EG exclusive nitrogen contains oxygen, while the XPS spectra of the carbonized samples of polyaniline-coated Si@EG show obvious N1s peaks. Therefore, by combining the TEM observation and the Raman analysis results of the carbonized samples, it can be confirmed that the polyaniline-coated Si@EG presents an outer N-doped carbon layer. The high-resolution spectra of Si 2p for both samples are shown in Figure 4c,f. There are two different peaks at 98.4 eV and 101.1 eV, which are due to the formation of the Si-Si bond and the Si-O bond. Furthermore, a small number of nano-silicon particles are exposed to oxygen in the process of preparing materials, resulting in a small amount of silicon oxide. This may also play a role in protecting active silicon from direct contact with electrolytes [33]. The high-resolution spectrum of C1s is shown in Figure 4b at 286.1 eV, which authenticates the existence of carbon in the composite sample, and it can be resolved into three single-handed component peaks. These component peaks contain C-C in aromatic rings at 284.8 eV, C-N or C=C at 283.7 eV and C-O at 282.3 eV. The high-resolution N1s spectrum (Figure 4d) at 397.3 and 399.5 eV is assigned to pyridine-type nitrogen (=N-) and pyrrole-type nitrogen (-NH-). These results are in good agreement with the other characterizations [34].

Through cyclic voltammetry (CV), the electrochemical performance of Si@EG@p-PANI is investigated. Figure 5a indicates the CV curves in the first five cycles at a scanning rate of 0.1 mV s^−1^ over a voltage range of 0.01–3 V (vs. Li^+^/Li). In the first cycle of cathodic scanning, a broad and weak cathodic peak appeared near 0.5 V, which may be attributed to the reaction between the electrode material and the electrolyte, as well as the formation of an irreversible SEI film on the electrode surface. In the subsequent cycles, the CV curve of the Si@EG@p-PANI composite exhibited three significant peaks at around 0.2 V in the reduction process after three cycles, corresponding to the lithiation of Si to amorphous α-Li_x_Si. Additionally, the two peaks at around 0.3 V and 0.5 V are attributed to the delithiation of the amorphous α-Li_x_Si to α-Si.

Figure 5b shows its constant current charging and discharging curve, with a voltage range of 0.01–3 V (vs. Li^+^/Li) and a current density of 100 mA g^−1^. There is an inclined potential plateau between 1.1 V and 0.2 V in the first discharge curve, which indicates the formation potential of SEI film. A long and flat voltage platform can also be seen below 0.2 V. There is also a voltage platform centered around 0.43 V on the charging curve, which corresponds to the process of silicon lithium removal. After several charge and discharge cycles, the subsequent discharge curves almost completely overlap, which also indicates that a stable electrochemical environment has been achieved. Si@EG@p-PANI has a discharge-specific capacity of 1491 mAh g^−1^ in the first cycle and one of 752 mAh g^−1^ after 100 cycles. In contrast, the Si@EG discharge-specific capacity decreased from 1942 mAh g^−1^ to 1100 mAh g^−1^ without p-PANI, and the pure silicon cathode rapidly decayed to 236 mAh g^−1^, which showed poor cycle stability (Figure 5c). The coulombic efficiency of Si@EG@p-PANI increased rapidly from the initial 78.3% to 96.5% in the second cycle, and the coulombic efficiency reached 99% in 100 cycles (Figure 5c). These results show that the nitrogen-doped carbon network structure formed by the pyrolysis of polyaniline can effectively improve the electrochemical cycle stability of Si@EG@p-PANI in the process of lithium/lithium removal, and they prove the mechanical stability of its structure, to some extent. After more than 100 cycles, its specific capacity will slowly show a trend of attenuation.

The magnifying power performance of Si@EG@p-PANI is shown in Figure 5d, with high specific capacity values of 1249, 980, 847 and 772 mAh g^−1^ at 0.1, 0.2, 0.5 and 1.0 A g^−1^, respectively. When the current density returns to 0.1 A g^−1^, the reversible capacity is maintained at values as high as 1050 mAh g^−1^, which indicates the favorable structural and conducting stability.

These results show that Si@EG@p−PANI composites not only have a good conductive network but also provide a good path for lithium-ion transmission. Polyaniline adheres well to the surface of exfoliated graphite by pyrolysis and establishes good electrical contact with it. In addition, the uniformly coated pyrolytic polyaniline carbon network structure also helps to form a stable SEI film on the electrode surface, and it can prevent electrolyte permeation and contact between the silicon and electrolyte. Silicon nanoparticles with a size of 100 nm are evenly distributed on the exfoliated graphite matrix, which can reduce the aggregation and pulverization of silicon nanoparticles, and this is very important in improving the cycle performance of the silicon-based anode. As a result, after 100 cycles, obvious cracks appeared on the surface of the pure silicon anode material. Moreover, Si@EG negative electrode material (Figure 6b) has a few cracks on its surface after 100 cycles. However, from Figure 6c,d, it can be seen that the surface of the Si@EG@p-PANI anode material still has a nearly smooth appearance after 100 cycles. It can also be concluded that the 3D sandwich structure of the Si@EG@p-PANI composite anode material has more cyclic stability.

## 4. Discussion

In summary, through simple pyrolysis and in situ polymerization, a 3D sandwich structure matrix encapsulated by silicon, exfoliated graphite and amorphous carbon core sheath nanostructures has been designed and successfully prepared for high-performance LiBs. The 3D multilayer graphite sandwich structure matrix provides high-conductivity channels for electrons and open channels for the rapid diffusion of electrolytes. In addition, the Si@EG composite remains bonded by the pyrolytic polyaniline carbon coating, which can effectively prevent the aggregation of silicon nanoparticles and provide more additional space and more reactive sites for adapting to volume changes. With the synergistic effect of pyrolytic polyaniline carbon coating and exfoliated graphite, this nanocomposite, as an anode for lithium-ion batteries, shows an excellent magnification performance, ultra-high specific capacity, excellent cycle stability and high conductivity. In addition, the summary of the capacity performance of different Si-based nanostructures reported in the literature and the comparative analysis compared with the results obtained in this work are shown in Table 1.

## Figures and Tables

**Figure 1 materials-16-01584-f001:**
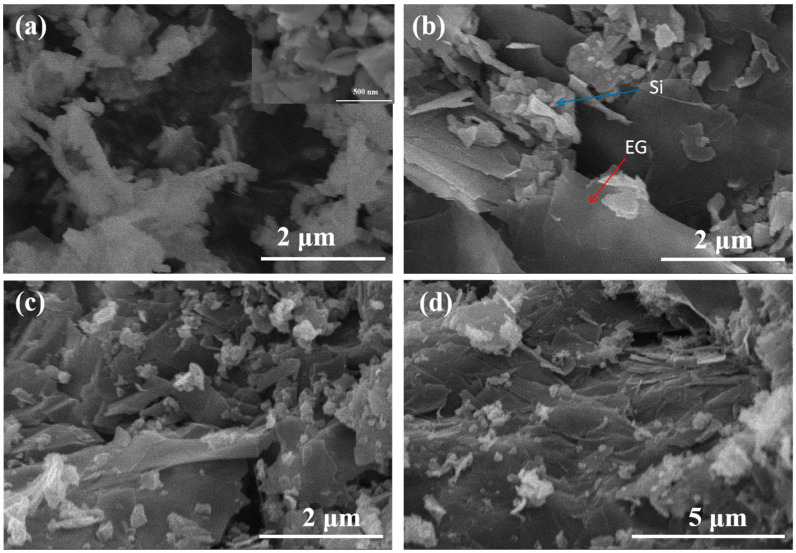
SEM images of (**a**) commercial nano-silicon powder, (**b**) synthesized Si@EG and (**c**,**d**) Si@EG@p-PANI composites.

**Figure 2 materials-16-01584-f002:**
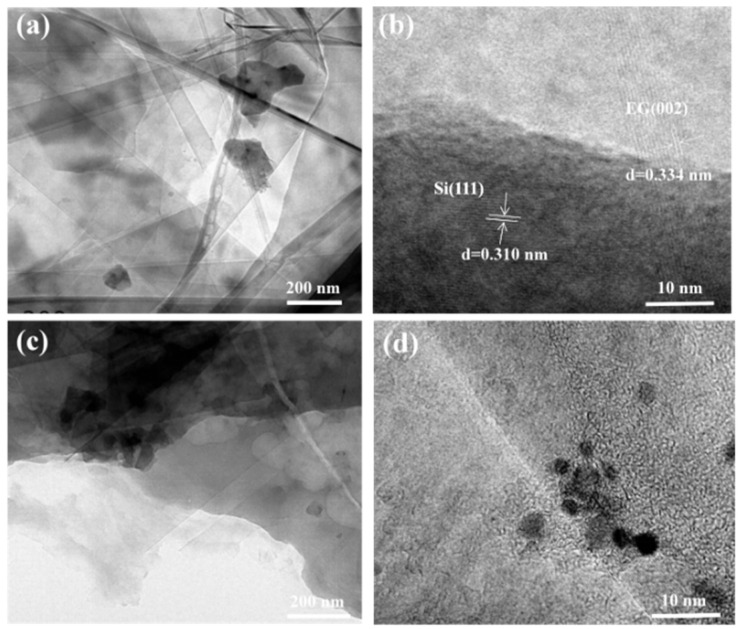
(**a**) TEM and (**b**) HRTEM images of Si@EG, (**c**) TEM and (**d**) HRTEM images of Si@EG@p-PANI composites.

**Figure 3 materials-16-01584-f003:**
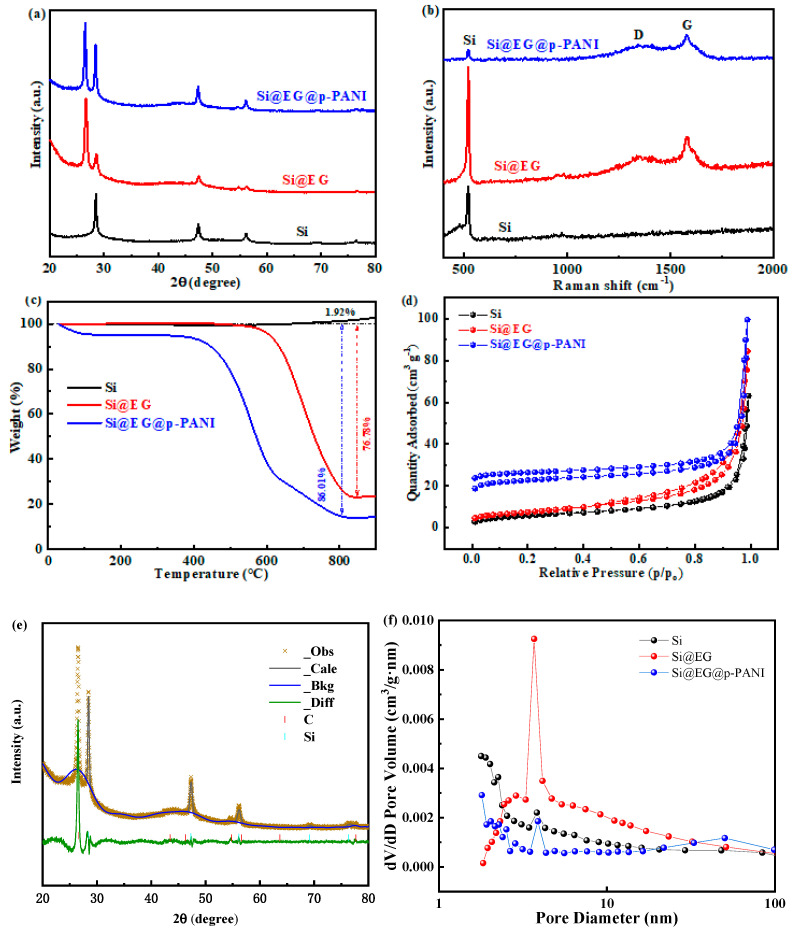
(**a**) XRD patterns, (**b**) Raman spectra, (**c**) TGA curves of raw Si powder, Si@EG composite and Si@EG@p−PANI composite in air gas at a heating rate of 5 °C min^−1^ and (**d**) N_2_ adsorption and desorption isotherm of raw Si powder, Si@EG composite and Si@EG@p−PANI composite. (**f**) Pore size distribution and (**e**) Rietveld refinement of XRD patterns.

**Figure 4 materials-16-01584-f004:**
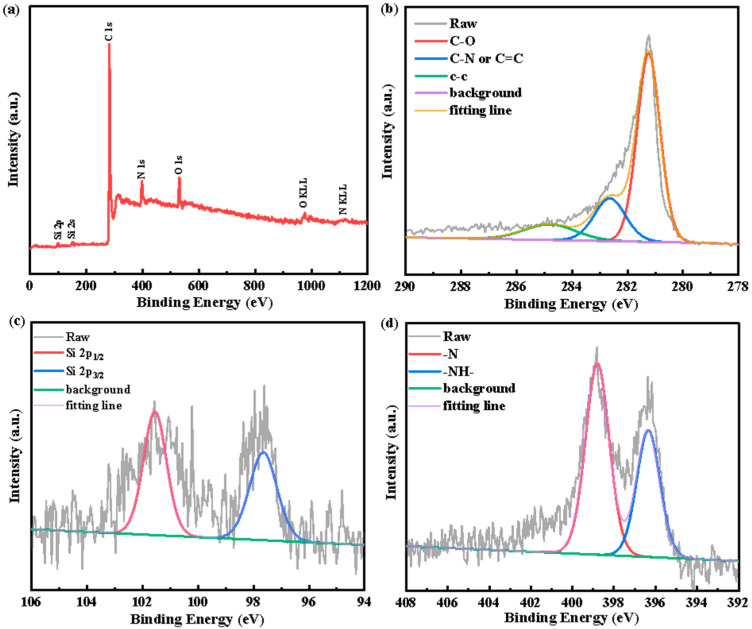
XPS spectra of the (**a**) Si@EG@p-PANI composite, (**b**) C1s signal, (**c**) Si2p signal, (**d**) N1s signal, (**e**) Si@EG composite, (**f**) Si2p signal, (**g**) C1s signal and (**h**) O1s signal.

**Figure 5 materials-16-01584-f005:**
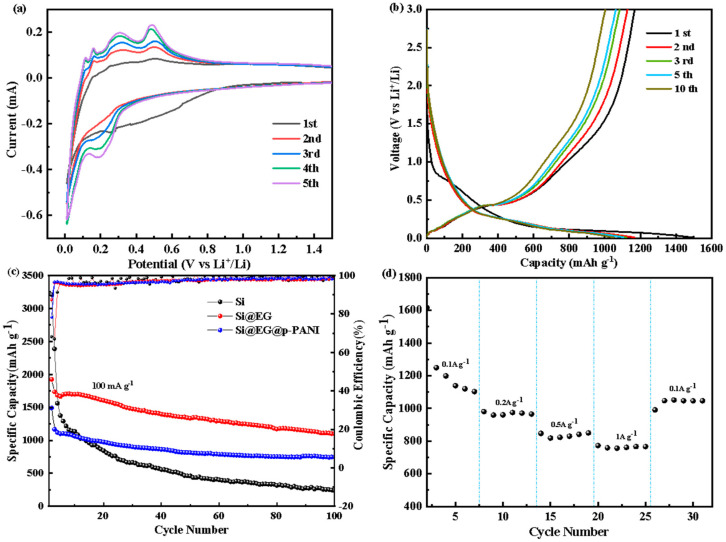
(**a**) Cyclic voltammetry curves of Si@EG@p−PANI from cycles 1 to 5 at a scanning rate of 0.1 mV s^−1^. (**b**) Galvanostatic charge−discharge profiles at a current density of 100 mA g^−1^ in the voltage range of 0.01−3 V. (**c**) Cycling performances and coulombic efficiency of Si@EG@p−PANI, Si@EG and nano-Si electrode at a current density of 100 mA g^−1^. (**d**) Rate capabilities of Si@EG@p−PANI at current densities from 0.1 to 1 A g^−1^.

**Figure 6 materials-16-01584-f006:**
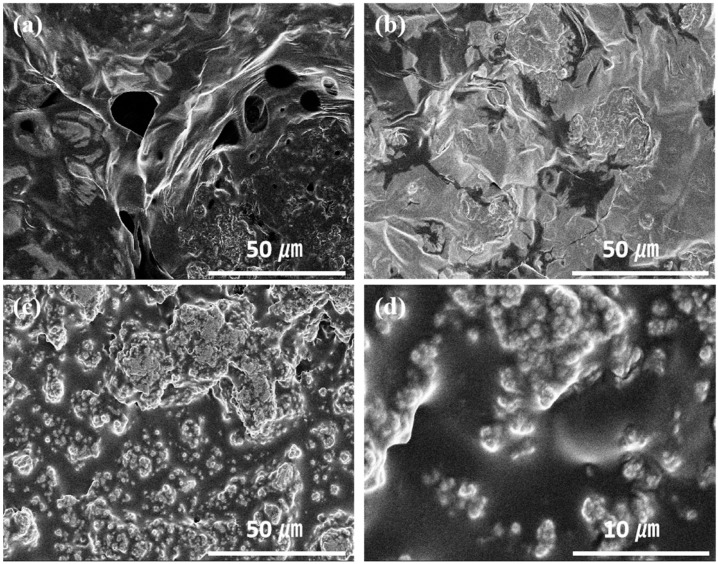
(**a**,**b**) SEM images of the surface of Si and Si@EG electrodes after the 100th cycle at a current density of 0.1 A g^−1^. (**c**,**d**) SEM images of the surface of Si@EG@p-PANI electrodes after the 100th cycle at a current density of 0.1 A g^−1^.

**Table 1 materials-16-01584-t001:** Comparison of the performance of different materials and methods.

Material	Method	Capacity [mAh g^−1^]	Reference
3D porous bulk Si particles	reduction of SiCl4 with sodium naphthalide	2800 (2 A g^−1^)	[9]
CN@P-Si	nitrogen-doped carbon coating	2000 (0.8 A g^−1^)	[13]
Si@C	in situ synthesized viaa facile one-pot solution	1120 (2 A g^−1^)	[17]
p-SiNPs@HC	hydrothermal	1400 (0.2 A g^−1^)	[21]
Si/Sn@C-G	ball milling and annealing process	612 (0.1 A g^−1^)	[16]

## Data Availability

Not applicable.

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
