# Peer review of "Nano-Silicon@Exfoliated Graphite/Pyrolytic Polyaniline Composite of a High-Performance Cathode for Lithium Storage"

_materials, 2023, doi:10.3390/ma16041584_

Round 1

Reviewer 1 Report

 In this paper, Si@EG composites and polyaniline were synthesized by liquid phase mixing, high-temperature solid phase method, and in-situ chemical polymerization of aniline monomer, respectively, to coat the surface of nano-silicon and exfoliated graphite composites (Si@EG). The structural and morphological characteristics were measured by SEM, TEM, XRD, Raman, TGA and BET techniques. Further, the electrochemical properties were measured and obtained a high initial specific capacity of 1491 mAh/g, but also maintained 752 mAh/g after 100 cycles at 100 mA/g.  The manuscript is suggested to be accepted after the following issues are addressed.

1)      In materials and method, there is no information about the materials source and purification etc. the authors should provide this information.

2)      The authors should add the Rietveld refinement of XRD data.

3)      The authors should add more information related to the XRD, such as crystalline size etc.

4)      The 100 cycles are too short for a practical device. Please describe the long cycling life of more than 100 cycles in the revised main text.

5)      The authors should add a comparative table or figure to compare these outcomes with the previous study

6)      Some related or latest studies should be discussed or cited, 10.1016/j.jssc.2022.123679; 10.1007/s10008-022-05305-9

7)      Many spelling and formatting typos in this paper, and we hope the authors can check and revise them thoroughly

Reviewer 2 Report

Interesting concept and results, but very problematic presentation. The text loses comprehensibility at various cases, with obvious (and very wrong) google-translations of terms and bad choices of sentence-openers/structures (the "Not only...but also..." structure was especially annoying and needless). The authors are requested to rewrite the whole manuscript for clarity and presentation, possibly resorting to editing services provided by the publisher or a 3rd party. 

The concept and title can be misleading. You are describing a process where you are coating the graphite with PANI and THEN pyrolyze it, to derive an amorphous N-doped carbon coating as deposited particles on the graphite. After this process, no PANI remains on the material, but yoy insist on refering to it in the title and elsewhere. You may use the term "PANI-derived carbon", but anything else is misleading.

Materials and Methods is incomplete and badly written.

Material characterization is very problematic as well. The authors are, as a rule, jumping to conclusions without a sound justification in a lot of cases. XRD, Raman, and BET/BJH are incomplete. TEM could be linked to XRD via SAED.

Electrochemical characterization was performed correctly and it was well written. All the rest need to be revisited. More and detailed remarks are given in the returned pdf. The authors are advised to correct and address ALL COMMENTS. 

References are, for some reason, at the greatest percentage, of Chinese articles. This is troubling. Please provide a greater variety of origin - there are a lot of articles that describe similar or relevant results from other origins. Please address this issue, as it is a cause of major ethical concerns.

Finally, the authors are advised to rescale all figures to the maximum size accomodated by the MDPI format, either full column or full page width is appropriate since you are using 4-figure panels.

Round 2

Reviewer 1 Report

Accept in present form.

Author Response

We would like to thank you for reading our paper carefully and giving the above positive comments. Thank you for your affirmation. We hope this article can be included as soon as possible.

Reviewer 2 Report

Almost none of the original comments were addressed.
